# Perceptions and knowledge of breast cancer and breast self-examination among young adult women in southwest Ethiopia: Application of the health belief model

Kenzudin Assfa Mossa  *

Department of Public Health, College of Medicine and Health Sciences, Wolkite University, Wolkite, Ethiopia

* kenzaheri@gmail.com

## Abstract

### Background

Breast cancer is still a recognized public health issue in Ethiopia. Despite this, the view-points and comprehensions of young women about the situation are unknown. Therefore, this study was carried out to assess the knowledge and perceptions of young adult women in Southwest Ethiopia about breast cancer and breast self-examination (BSE).

### Methods

A community-based cross-sectional study was carried out in the Gurage zone, southwest Ethiopia, in 2021. A total of 392 young adult women were randomly selected from both urban and rural strata using a three-stage stratified sampling process. A pretested question-naire was used to collect the data. For data entry, Epi-data 4.6 with a double-entry approach was used, and for analysis, SPSS 26 was used. Bivariate and multivariable logistic regression analyses were performed to identify variables associated with BSE behavior. A p-value of 0.05 or below was considered statistically significant with a 95% CI.

### Results

The respondents' ages ranged from 20 to 24, with a mean of 21.25 (±1.32) years. Breast cancer and BSE were unknown to more than 80% of the study participants. A large proportion of young adult women had low perceived susceptibility (97.6%), low threat of breast cancer (96%), and low self-efficacy to perform BSE (91.4%). BSE was conducted by 23.1% of the participants occasionally. Being married (AOR = 5.31, 95% CI = 2.19–12.90), having good outcome expectations of BSE (AOR = 2.05, 95% CI = 1.16–3.61), having good BSE knowledge (AOR = 1.22, 95% CI = 1.04–1.45), having high perceived susceptibility (AOR = 1.12, 95% CI = 1.05–1.20), high perceived severity (AOR = 1.78, 95% CI = 1.02–3.09), and having high self-efficacy to do BSE (AOR = 1.05, 95% CI = 1.01–1.09) were all significant predictors of BSE practice.

**Data Availability Statement:** The minimal data set is fully available on Figshare (DOI: 10.6084/m9.figshare.20103206).

**Funding:** The author(s) received no specific funding for this work.

**Competing interests:** The authors have declared that no competing interests exist.

## Conclusions

Young adult women were less concerned about breast cancer and had insufficient knowledge of breast cancer and breast self-examination. They have little knowledge of, confidence in, or experience with BSE. The practice of BSE was associated with increased perceived susceptibility, self-efficacy, severity, outcome expectations, and BSE knowledge. Therefore, these variables should be considered when developing educational interventions for young women.

## Introduction

Breast cancer has surpassed lung cancer as the most common malignancy among women, with 2.3 million new cases and 685, 000 deaths recorded in 2020 [1]. Every year, it kills approximately 500,000 women, and disproportionate deaths occurs in low-resource countries [2].

In Ethiopia, breast cancer is widely recognized as a serious public health concern and a priority cancer for intervention [3,4]. Approximately 10,000 Ethiopian women are affected. 90% of women are diagnosed with advanced breast cancer, with tens of thousands more cases undetected [4].

Many people believe that breast cancer only affects women in their forties and fifties [5,6]. Breast cancer, on the other hand, can and does affect young women. Every year, about 13,000 women under 40 are diagnosed with breast cancer. More than 80% of young women find out about their abnormal breast condition on their own. The global increase in breast cancer incidence is evident in all age categories, with women under 50 experiencing the greatest increase [7,8]. The number of young women with risk factors for breast cancer has been shown to be alarmingly high [9]. Young women with cancer are often diagnosed with advanced stage breast cancer. They have a variety of issues, including a higher likelihood of biologically aggressive illness and metastatic disease at diagnosis, which leads to a worse prognosis, more aggressive treatments and long-term treatment-related toxicities, and unique psychosocial concerns [10–12]. As a result, their survival rate is lower than older women [13].

The impact of early detection of breast cancer on morbidity and mortality is significant. If this cancer was detected early, a 95% chance of survival could be achieved [8,14]. The American Cancer Society recommends three screening options: mammography, clinical breast examination (CBE), and breast self-examination (BSE). Despite the fact that mammography is the most effective tool for early breast cancer detection, it is prohibitively expensive in developing nations [8].

When a woman palpates (feels) and inspects her own breasts for lumps, abnormalities, or swelling, this is known as breast self-examination [14]. Although there is debate about the efficacy of BSE in countries where mammography and CBE are widely available, BSE remains a simple, non-invasive, cost-effective, and accessible method of early detection in developing countries like Ethiopia, where mammography is extremely difficult to obtain [5,8,13,15].

It is critical for women to understand their own breasts and be mindful of any changes. Women can be empowered by BSE and other health practices that allow them to take control and responsibility for their own health [13,15].

BSE is still recommended as a general method for increasing breast health awareness, which makes it easier for women to detect any changes that may arise [8,16,17]. Despite the benefits, many women in many countries do not perform regular self-breast checks

[8,13,15,18]. Many women cited a lack of competence as the primary reason for not doing BSE [16,17,19]. For younger women, BSE education and adherence are a doorway to health-promoting behaviors, laying the groundwork for eventual adherence to clinical breast examination and mammography screening, if available [13,15]. Breast cancer fundamentals, such as risk factors, early warning signs, and screening modalities, such as BSE, should be taught to all women starting at a young age [8,9,18,20,21]. To promote breast self-examination practice, their concern and confidence should be improved [8,22].

Model-based educational interventions are more effective for the BSE screening behavior of women [23]. Interventions that target the threat of breast cancer and the benefits of breast self-examination may help improve knowledge and skills for performing breast self-examinations [24].

However, little is known about Ethiopian young adult women's perceptions of breast cancer and BSE. Therefore, the current research attempted to determine how young adult women in Southwest Ethiopia perceive and understand breast cancer and BSE. The findings may be useful in national and local breast health promotion efforts.

## Material and methods

### Study area and study design

The study was conducted in the Guraghe zone, southwest Ethiopia, from June 15 to September 30, 2021. The zone is one of the zonal administrations in the regional state in southern Ethiopia. Its administrative center, "Wolkite," is located 158 kilometers away from Addis Ababa. It was divided into five town administrations and sixteen districts, which were further divided into 423 (403 rural and 20 urban) smaller administrative units, "kebeles" and 88,587 households. In 2021, the zone had a population of 1,807,892 people, of whom 43.8% were urban dwellers. Young adult women accounted for 7.6% of the total, while non-pregnant reproductive-age (15–49 years old) women accounted for 19.8%. The zone has eight hospitals (1 referral, 1 general, and 6 primaries), 75 health centers, and 416 health posts, all of which provide various levels of health care to the public [25].

To investigate views and understanding of breast cancer and BSE, a community-based cross-sectional study was done on randomly selected 392 young adult women.

### Study population and eligibility criteria

All young adult women (aged 20 to 24) living in the Guraghe zone were the source population, while those who were permanent residents in selected districts were the study population for this study. The study included young adult women who were available throughout the data collection period, consented to participate in the study, and completed all the questions. Those who refused to participate or were critically ill to interact with data collectors were excluded.

### Sample size determination and sampling procedure

The sample size required for this study was determined by using the formula to estimate a single population proportion, assuming 20.3% of young adult women have ever practiced BSE [8], 95% CI, 5% margin of error, at alpha 0.05.

$$n = \frac{(z\frac{\alpha}{2})^2 * p(1-p)}{d^2} \quad n = \frac{(1.96)^2 * 0.203(0.797)}{0.05^2} = 249$$

The formula produced n = 249. After adding 5% for non-response, and multiplying by 1.5 for design effect, the final calculated sample size was 392 young adult women.

The study participants were selected using a three-stage stratified sampling technique from urban and rural strata. In the first stage, three town administrations and six districts were randomly selected. 4 kebeles from each of the 3 selected town administrations and 48 kebeles from each of the 6 selected districts were randomly picked in the second stage of the sampling process. The sample size was proportionally assigned to each kebele, depending on the number of households eligible to participate in each kebele. A census survey was conducted to determine eligible households. In the third stage of the selection process, households containing young adult women were recruited using a systematic random sampling technique. A sampling interval was determined for each kebele, and the first dwelling was randomly selected. When there were multiple eligible participants in a house, a lottery method was employed to choose one. The data were collected by 25 trained health extension workers under the supervision of five health officers and nurses. Households that were either closed or lacked a respondent were visited three times. Those who were unavailable on the third (last) visit were listed as non-respondents.

## Data collection tools and measurements

A pretested questionnaire was used as a data collection instrument, adapted from literature [13,26–30]. It was organized into four sections: sociodemographic, knowledge, perception, and practice of BSE. Participants fill out questionnaires in the presence of data collectors. when someone couldn't read due to visual difficulties or illiteracy interviewer-administered questions, which had the same wording and form as self-administered questionnaires were used. Using the American Cancer society guidelines [29], the knowledge of breast cancer warning signs and awareness of breast cancer risk factors were assessed using 11 questions with "yes" or "no" response and 10 items with a "true or false" response option, respectively. Five multiple-choice questions adapted from prior research were used to test knowledge of breast self-examination [13]. This form detailed the proper age to begin BSE, the best time to perform BSE, the frequency of BSE, and the BSE procedure.

Correct responses were coded as 1, while incorrect responses were coded as 0. Each correct response was added together to create a composite score. The greatest possible scores for knowledge of risk factors, warning signs, and BSE were 10, 11, and 5 respectively. The higher the score indicated, the better knowledge. The Cronbach's alpha coefficient for the internal consistency reliability of warning signs, BC risk factors, and measurement of BSE knowledge was 0.788, 0.572, and 0.635, respectively. A revised Bloom's cut-off of 80% was used to determine adequate knowledge [30].

The revised Champion Health Belief Model Scale was used to assess respondents' perceptions. It's a widely used tool for analyzing variables like perceived susceptibility, severity, perceived barriers, benefits, and self-efficacy. According to previous literature, the reliability coefficient for each subscale calculated using Cronbach's alpha ranged from 0.58 to 0.91 [26–28]. On a five-point scale, participants were asked to rate each item, ranging from 1 strongly disagree to 5 strongly agree. The corresponding scores were added together, with values ranging from 7 to 35 for susceptibility, 8 to 40 for severity, 7 to 35 for benefits, 7 to 35 for barriers, and 10 to 50 for self-efficacy. Higher scores were expected to suggest a more positive attitude toward breast cancer and BSE, with the exception of BSE barriers.

Respondents' perceived threat was calculated as the sum of susceptibility and severity scores. By subtracting the perceived barrier score from the perceived benefits, the perceived outcome expectation was computed. The mean value served as the cutoff point for dichotomizing each subscale. In this study, Cronbach's alpha values were calculated as 0.72 for

susceptibility, 0.75 for severity, 0.77 for BSE benefits, 0.63 for BSE barriers, and 0.79 for BSE self-efficacy.

Respondents under investigation were also asked about the practice of their breast self-examination for breast cancer. Those who answered "Yes" were considered performed the behavior and were coded as 1. Three more inquiring questions were then used to examine the appropriateness of the examination time and method. Monthly self-examination with three middle finger pads a week after each menstruation was considered good breast self-examination practice in this study [13].

## Data processing and analysis

The data were double-checked for accuracy and consistency. For data entry and analysis, EPI-data version 4.6 software package and the Statistical Package for Social Sciences (SPSS) version 26 software package were used. Frequency, tables, figures, mean, and standard deviation (SD) were used to present the descriptive data. Variables with a p-value <0.25 in the bivariate binary logistic regression analysis were selected to fit the final model. Variance inflation factor (VIF) of >10 and tolerance <0.1 was considered suggestive of multi-collinearity. However, no multi-collinearity was detected during the analysis. Multivariate logistic regression analysis was performed to control the potential effect of confounders and identify the major factors influencing breast self-examination practice. The model's fitness was assessed using the Hosmer and Lemeshow goodness-of-fit test (p-value = 0.301). Finally, the Adjusted Odds Ratio (AOR) with 95% CI was used to evaluate the strength of the association between the explanatory and outcome variables. Independent variables with a P-value < 0.05 were declared to have a statistically significant association with the outcome variable after controlling for the effects of confounders.

## Data quality control

The questionnaire was pilot tested on 5% of the sample in an unselected district one week before the actual data collection period to ensure data quality. The research objectives, data collection processes, and interview protocols were all explained to data collectors and supervisors. All the data collection processes were closely overseen, and the completeness of collected data was checked daily by the lead investigator and supervisors. The information was properly coded, and a double-entry strategy was applied.

## Ethical clearance and consent for participation

Ethical approval was obtained from Wolkite University's institutional ethical review board. The purpose of the study was communicated to the respondents, and signed consent from each participant was obtained prior to data collection.

## Results

### Socio-demographic characteristics and sources of information

A total of 373 young adult women responded to the questionnaire, with a rate of 95.15% response. The participants' ages ranged from 20 to 24, with a mean of 21.25(±1.32) years. The majority of the study's participants were single (92.23%) and rural dwellers (62.73%).

About half (54.42%) of them attended primary to secondary school. Only 50.13% had heard of breast self-examination before. The media was cited as the key source of information by nearly sixty percent (57.91%) of young adult women. Fifteen (4.04%) of respondents had a positive family history (Table 1).

**Table 1. Socio-demographic profile of young adult women Gurage zone.**

| Variables | Category | Frequency | Percent (%) |
|---|---|---|---|
| Mean age = 21.25 (±1.32) years. | | | |
| Resident area | Rural | 234 | 62.73 |
| | Urban | 139 | 37.27 |
| Educational Background | Illiterate | 9 | 2.41 |
| | Primary to Secondary | 203 | 54.42 |
| | College and above | 161 | 43 |
| Marital status | Single | 344 | 92.23 |
| | Married | 29 | 7.77 |
| Ever heard about breast cancer | Yes | 370 | 99.1 |
| | No | 3 | 0.80 |
| Heard about BSE | Yes | 187 | 50.13 |
| | No | 186 | 49.87 |
| Sources of information | Media/TV/radio | 216 | 57.91 |
| | Health workers | 107 | 28.69 |
| | Friends | 42 | 11.26 |
| | Not remembered | 8 | 2.14 |
| Had family history of BC | Yes | 15 | 4.04 |
| | No | 358 | 95.98 |

### Knowledge of risk factors for breast cancer

In terms of respondents' knowledge of breast cancer risk factors, alcohol drinking (49.06%) and cigarette smoking (47.72%) were the two most well-known risk factors. Only a small percentage of respondents knew that a high-fat diet (34.58%), oral contraceptive pill use (31.37%), late menopause (28.15%), having their first child at a late age of 30 years (29.22%), family history (29.76%), early onset of menarche (19.30%), and advanced age (25.74%) are risk factors for developing breast cancer. Overall, 305 young adult women (81.77%) showed low risk factor awareness, while only 55 (14.70%) and 13 (3.50%) had moderate and high knowledge about risk factor, respectively (Table 2).

### Knowledge of early warning signs of breast cancer

Around three-quarters of study participants reported soreness in the breast (76.41%), pain in the breast region (77.48%), and ulceration (74%) as breast cancer warning signs. Discoloration of the breast (65.68%), painless breast lump (66.22%), change in breast size (62.73%), dimpling of breast skin (60.86%), and dry skin (60.32%) were the next most reported warning signs, followed by a nipple discharge (58.45%) and a lump under the armpit (50.40%). Less than half (40.75%) of the participating young women knew that weight loss is a warning sign for breast cancer. In this study, 37.53%, 21.72%, and 40.75% of respondents had good, moderate, and low warning sign knowledge, respectively (Table 3).

When it came to overall BC knowledge, nearly three-quarters of young adult women (73.70%) had inadequate information (Fig 1).

### Knowledge and practice of breast self-examination

In this study, only 86 (23.06%) and 7 (1.88%) of young adult women had conducted BSE at least once and regularly, respectively. In terms of BSE knowledge, 79.89% of respondents had poor knowledge of BSE. The optimal time for BSE and frequency is unknown to 58.71% and

**Table 2. Knowledge of young adult women on breast cancer risk factors Gurage zone.**

| Variables | Category | Frequency | Percent (%) |
|---|---|---|---|
| Alcohol consumption | Yes | 183 | 49.06 |
| | No | 190 | 50.94 |
| Cigarette smoking | Yes | 178 | 47.72 |
| | No | 195 | 52.28 |
| Consumption of high-fat diet | Yes | 129 | 34.58 |
| | No | 244 | 65.42 |
| Oral contraceptive pills Use | Yes | 117 | 31.37 |
| | No | 256 | 68.63 |
| Late menopause (after the age of 55 years) | Yes | 105 | 28.15 |
| | No | 268 | 71.85 |
| First child at the late age of 30 years | Yes | 109 | 29.22 |
| | No | 264 | 70.78 |
| Family history | Yes | 111 | 29.76 |
| | No | 262 | 70.24 |
| Early-onset of menarche before 12 years | Yes | 72 | 19.30 |
| | No | 301 | 80.70 |
| Advanced age | Yes | 96 | 25.74 |
| | No | 277 | 74.26 |
| Overall knowledge about breast cancer risk factors | Low | 305 | 81.77 |
| | Moderate | 55 | 14.70 |
| | High | 13 | 3.50 |

65.68% of subjects, respectively. More than seventy percent of them (71.05%) have no idea how to do BSE (Table 4).

## Perception towards breast cancer and BSE

This study showed that 97.59% of young women believed they were not susceptible to BC, and their perceived severity score was high (54.16%). Two hundred fifty-four (68.10%) of respondents had a high perceived benefit score on breast self-examination, while 269 (72.12%) had low perceived confidence or self-efficacy to performing BSE. The vast majority of young women (95.98%) were less concerned about breast cancer, and over half (59.79%) had low expectations regarding the outcome or net benefit of BSE. When it comes to the barriers to practice BSE, more than ninety percent of young adult women (91.42%) have low perceived barriers (Table 5).

## Factors associated with breast self-examination

According to the multivariate logistic regression analysis, married young adult women were 5.31 times more likely than unmarried young adult women to perform BSE (AOR = 5.31, 95% CI: (2.19–12.90). BSE knowledge was significantly associated with BSE practice. After all the other factors were kept constant, per one-unit increase in knowledge of BSE, the odds of practicing breast self-examination increased by 1.22 times (AOR = 1.22, 95% CI: 1.04–1.45). For each unit increase in perceived susceptibility to breast cancer and perceived self-efficacy to do BSE, the odds of performing BSE increased by 1.12 (AOR = 1.12, 95% CI:(1.05–1.20) and 1.05 (AOR = 1.05, 95% CI: (1.01–1.09), respectively. The study found that young adult women with a high perceived severity score and positive outcome expectations for BSE were 1.78

**Table 3. Respondents knowledge about warning signs of breast cancer, Gurage zone.**

| Warning sign | Responses | Frequency | Percent (%) |
|---|---|---|---|
| Mean ± SD = 7.18 ±2.82 out of 11 | | | |
| Overall warning sign knowledge | High | 140 | 37.53 |
| | Moderate | 81 | 21.72 |
| | Low | 152 | 40.75 |
| Painless breast lump | Wrong | 126 | 33.78 |
| | Correct | 247 | 66.22 |
| Lump under armpit | Wrong | 185 | 49.59 |
| | Correct | 188 | 50.40 |
| Nipple discharge | wrong | 155 | 41.55 |
| | correct | 218 | 58.45 |
| Change in breast size | wrong | 139 | 37.27 |
| | correct | 234 | 62.73 |
| Pain in breast region | wrong | 84 | 22.52 |
| | correct | 289 | 77.48 |
| Dimpling of breast skin | wrong | 146 | 39.14 |
| | correct | 227 | 60.86 |
| Dry skin in nipple region | wrong | 148 | 39.68 |
| | correct | 225 | 60.32 |
| Weight loss | wrong | 221 | 59.25 |
| | correct | 152 | 40.75 |
| Pain or soreness in the breast | wrong | 88 | 23.59 |
| | correct | 285 | 76.41 |
| Discoloration /dimpling of the breast | wrong | 128 | 34.32 |
| | correct | 245 | 65.68 |
| Ulceration of the breast | wrong | 97 | 26 |
| | correct | 276 | 74 |

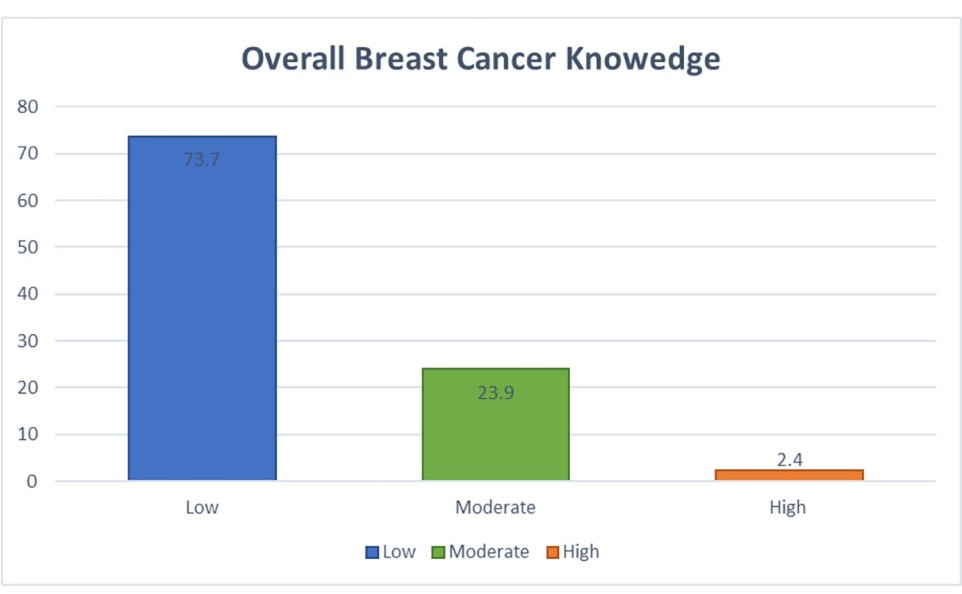

**Fig 1. Young adult women's knowledge of breast cancer Gurage zone.**

**Table 4. Young adult women's BSE knowledge and practice, Gurage Zone.**

| Knowledge of and practice of BSE | Category | Frequency (n) | Percent (%) |
|---|---|---|---|
| Have you ever done BSE | Yes | 86 | 23.06 |
| | No | 287 | 76.94 |
| Do you know how to perform BSE | Yes | 146 | 39.14 |
| | No | 227 | 60.86 |
| Can you successfully do BSE | Yes | 127 | 34.05 |
| | No | 246 | 65.95 |
| Knowledge about the frequency of BSE (n = 373) | Wrong response | 245 | 65.68 |
| | Correct response | 128 | 34.31 |
| Knowledge about appropriate time for BSE (n = 373) | Wrong response | 219 | 58.71 |
| | Correct response | 154 | 41.29 |
| Knowledge about BSE procedure (n = 373) | Wrong response | 265 | 71.05 |
| | Correct response | 108 | 28.95 |
| Overall BSE knowledge (n = 373) | Poor knowledge | 298 | 79.89 |
| | Good knowledge | 75 | 20.11 |

(AOR = 1.78, 95% CI: (1.02–3.09) and 2.05 (AOR = 2.05, 95% CI: (1.16–3.61) times more likely than their counterparts to engage in BSE behavior, respectively (Table 6).

## Discussion

Young adult women are an important target group for breast health promotion [13,15]. Breast self-examination should be done beginning the age of 20. Therefore, it's critical for women to understand their own breasts and be aware of any changes [13]. However, little study has been done in Ethiopia on young adult women's perceptions and comprehension of breast cancer and BSE. The purpose of this study was to find out what young adult women in Southwest Ethiopia thought about breast cancer and BSE. A full grasp of these key concepts is considered crucial for developing a successful public health intervention to increase women's awareness and concern about their breast health [16,20].

**Table 5. Perception of young adult women towards breast cancer and BSE Gurage Zone.**

| Variable | Response category | Frequency | Percent (%) |
|---|---|---|---|
| perceived susceptibility | Low | 364 | 97.59 |
| | high | 9 | 2.41 |
| Perceived severity | Low | 171 | 45.84 |
| | high | 202 | 54.16 |
| Perceived threat | Low | 358 | 95.98 |
| | high | 15 | 4.02 |
| Perceived benefit | Low | 119 | 31.90 |
| | high | 254 | 68.10 |
| Perceived barrier | Low | 341 | 91.42 |
| | high | 32 | 8.58 |
| Perceived self-efficacy | Low | 269 | 72.12 |
| | high | 104 | 27.88 |
| Perceived outcome expectation | Negative | 223 | 59.79 |
| | Positive | 150 | 40.21 |

**Table 6. Factors associated with breast self-examination based on multivariate logistic regression analysis among young adult women Gurage zone.**

|  | Category | BSE practice | | COR 95% CL | AOR (95%CL) |
|---|---|---|---|---|---|
|  |  | Yes | No |  |  |
| Marital status | Married | 16 | 13 | 4.82(2.21–10.48) * | 5.31(2.19–12.90) ** |
|  | Single | 70 | 274 | 1 | 1 |
| Family History of BC | Yes | 8 | 7 | 4.10(1.44–11.67) ** | 2.23 (0.63–7.89) |
|  | No | 78 | 280 | 1 | 1 |
| Severity | high | 55 | 147 | 1.69 (1.03–2.78) ** | 1.78 (1.02–3.09) * |
|  | low | 31 | 140 | 1 | 1 |
| BSE knowledge [a] |  |  |  | 1.38(1.19–1.60) * | 1.22(1.04–1.45) ** |
| Susceptibility [a] |  |  |  | 1.14(1.08–1.20) * | 1.12(1.05–1.20) * |
| Barriers [a] |  |  |  | 0.93(0.88–0.99) ** | 0.94(0.88–1.01) |
| Self-efficacy [a] |  |  |  | 1.07(1.03–1.12) * | 1.05(1.01–1.09** |
| Outcome expectations | Positive | 46 | 104 | 2.02(1.24–3.29) ** | 2.05(1.16–3.61) ** |
|  | Negative | 40 | 183 | 1 | 1 |

** p< = 0.05

*p-value <0.001, Hosmer and Lemeshow goodness-of-fit test (p = 0.301)

[a] continuous variables.

In the current study, approximately half (49.87%) of young adult women had never heard of breast self-examination. Similarly, 47.5% of young college students in Addis [31] and 62.1% of female high school students in Turkey [13], and 26.5% of young women in Cameroon [18] had never heard of BSE. This data implies young women are less informed about breast self-examination and breast health in general. It is possible that this is due to the fact that cancer receives less attention [3]. Breast health should be given sufficient attention and included in school curricula so that all women are informed about it from an early age [8,9,18,20,21].

More than half of our respondents mentioned the media as their primary source of information on breast cancer and BSE, which is consistent with prior studies [13,31]. These findings reveal the significance of media outlets, such as audio-visuals and social media platforms, in disseminating breast health-related information to the general population.

Congruent with other studies [13,32,33]. a great proportion (81.77%) of the present study participants had poor knowledge of breast cancer risk factors. Almost three-fourths (73.70%) of young adult women had low knowledge of breast cancer. Similarly, 75.90% of female students in Mekelle University [34] and 56.20% in Sudan Khartoum had poor knowledge of breast cancer [35]. Nearly 38% of respondents have excellent warning sign knowledge, whereas 138 (40.75%), on the other hand, have insufficient warning sign knowledge. On the other hand, the knowledge of warning signs (72.80%) in Lebanese women in Beirut was the highest proportion [36].Since older and married women are more interested in and aware of BC [20,37,38], this difference could be explained by our younger participants (20–24 years of age), who were younger than those in Beirut (18–65 years), and whose proportion of married people (7.77%) was significantly lower than Beirut's (49.5%) [36].

Breast self-examination knowledge was found to be quite low in this study. More than three quarters (79.89%) of young women had poor knowledge of BSE, and 71.05% of them were unfamiliar with the process. Moreover, 58.71% and 65.68% of them were unaware of the right timing and frequency of breast self-examination, respectively. This finding was consistent with the study conducted in Turkey, which showed that 66%, 75.4%, and 65.4% of high school female students were unaware of the frequency of BSE, the appropriate time for BSE, and the BSE procedure, respectively [13]. The data demonstrates that BSE is a lesser-known practice

that requires extensive public involvement and attention to improve uptake. It could be because of socioeconomic disparities in access to health education [37] the Low living standards, a lack of health awareness, and less societal visibility and attention paid to BSE may all contribute to their lack of in-depth understanding of BC risk factors and BSE [3,14,38].

According to the current study, there is a strong link between BSE knowledge and BSE practice. In keeping with this, it was noted that the likelihood of performing BSE was much higher among those who had better knowledge of BSE than their counterparts [13]. Studies conducted among undergraduate female students in Cameroon [18] substantiate the finding.

In this study, only 23.06% and 1.88% of young adult women performed breast self-examination occasionally and regularly, respectively. In line with this conclusion, 20% and 6.7% of female high school students in Turkey [13], and 24.1% and 8.1% of female secondary school students in Ghana, respectively, performed BSE regularly and occasionally, as indicated [8]. However, in Sri Lanka's Colombo [15], a lesser percentage (6.17%) of adolescent girls had ever performed it. This lower practice of BSE could be due to lack of knowledge about the BSE procedure. Because 71% of our participants were unaware of it. In previous research, not knowing how to perform BSE was the main reason for not doing so [19,34]. These findings underscore the importance of providing young women with accurate breast health information to promote self-examination behavior. Participants' demographic characteristics, such as age and marital status, could potentially be associated to their lack of BSE knowledge and experience. In this study 92.23% of young adult women were single and married young adult women were 5.31 times more likely than unmarried young adult women to perform BSE (AOR = 5.31, 95% CI: (2.19–12.90). Research undertaken in southern Ethiopia and Malaysia [19,39] supports this finding. In other words, unmarried women may face more perceived barriers to breast screening [40]. This could also be due to the fact that married women receive greater social support and are exposed to maternal health care and related information during antenatal care, delivery, and the postpartum period.

The majority of young adult women had reduced perceived susceptibility (97.59%), low self-efficacy (72.12%), and severity (45.84%). According to the current study, a high proportion of study participants have low perceived obstacles (91.42%). This was higher than the findings of a study conducted in Adwa town [22], which indicated 47% low perceived susceptibility, 58% low barrier, and 38% low self-efficacy. In the Philippines, female teachers had a low perception of their susceptibility (9%) and a low degree of confidence (33%) in executing BSE [16].

Overall, our study revealed that 95.98% of young adult women were less concerned about breast cancer, and 59.79% had a lower expectation or net benefit from BSE. Because our respondents were substantially younger, it is possible they felt less threatened and had lower expectations.

The results of a multivariate logistic regression analysis revealed that the perceived susceptibility score of young adult women under study was substantially related to breast self-examination practice. This finding is consistent with studies from Ethiopia [22,24], which revealed that high personal breast cancer susceptibility increased the likelihood of BSE.

Furthermore, young women's confidence was strongly associated with their breast self-examination performance. Other researchers corroborated this finding [19,22,41]. Knowledge builds confidence. More successful BSE could result from interventions aimed at enhancing women's confidence in their ability to do BSE, as well as initiatives to improve breast cancer awareness.

Our study showed that young adult women with positive expected outcomes were more than twice as likely to perform BSE as their counterparts. This claim backed up a study conducted in Ethiopia [24]. As a result, educating women about the benefits of BSE and reducing

barriers such as lack of knowledge about BSE could be an intervention area for improving BSE practice.

Consistent with a study done in Iran [20], the current study found that respondents' perceived benefits and perceived barriers scores were not statistically associated with actual BSE practice. This conclusion, however, contradicts the findings of Iranian studies [21,41], which found that high perceived benefit and lower perceived barriers were better predictors of BSE. It is possible the disparity is due to the difference in study population and study period.

## The strengths and limitations of the study

The application of the revised champions' health belief model as a theoretical framework can be the strength of this research work. Since only young adult women participated in the study, it does not reflect the perceptions and knowledge of young adult males. Moreover, as a cross-sectional study, the results cannot demonstrate the causal relationship between dependent and independent variables.

## Conclusions

Generally, the findings of this study revealed that the knowledge, threat, and performance of breast self-examination were very low. The likelihood of practicing BSE was higher in young adult women who had exhibited higher susceptibility, severity, self-efficacy, positive outcome expectations, and good BSE knowledge. Therefore, interventions targeting young females should be devised based on these variables, to improve the rates of regular breast self-examination.

## Acknowledgments

First and foremost, our heartfelt gratitude goes to the Silte zone health department for its support in carrying out this study.

## Author Contributions

**Conceptualization:** Kenzudin Assfa Mossa.

**Formal analysis:** Kenzudin Assfa Mossa.

**Methodology:** Kenzudin Assfa Mossa.

**Supervision:** Kenzudin Assfa Mossa.

**Writing – original draft:** Kenzudin Assfa Mossa.

**Writing – review & editing:** Kenzudin Assfa Mossa.

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
