## [Decision Letter · Decision Letter 0]

9 May 2022

PONE-D-22-07638Perceptions and knowledge of breast cancer and breast self-examination among young adult Ethiopian women: application of the health belief modelPLOS ONE

Dear Dr. Assfa,

Thank you for submitting your manuscript to PLOS ONE. After careful consideration, we feel that it has merit but does not fully meet PLOS ONE’s publication criteria as it currently stands. Therefore, we invite you to submit a revised version of the manuscript that addresses the points raised during the review process.

We look forward to receiving your revised manuscript.

Kind regards,

Alvaro Galli

Academic Editor

PLOS ONE

Journal Requirements:

Reviewers' comments:

Reviewer's Responses to Questions

**Comments to the Author**

1. Is the manuscript technically sound, and do the data support the conclusions?

Reviewer #1: Partly

Reviewer #2: Yes

2. Has the statistical analysis been performed appropriately and rigorously? 

Reviewer #1: Yes

Reviewer #2: Yes

3. Have the authors made all data underlying the findings in their manuscript fully available?

Reviewer #1: No

Reviewer #2: No

4. Is the manuscript presented in an intelligible fashion and written in standard English?

Reviewer #1: Yes

Reviewer #2: Yes

5. Review Comments to the Author

Reviewer #1: Perceptions and knowledge of breast cancer and breast self-examination among young adult Ethiopian women: application of the health belief model .

REVIEW COMMENTS

The study was carried out to assess the knowledge and perceptions of young adult women in Southwest Ethiopia about breast cancer and breast self-examination (BSE). This is a good manuscript with very robust methods. The methodology has been comprehensibly and logically described for reproducibility. The write up and presentation good, however, some concerns need to be addressed. I recommend a major revision

Major Revision

Abstract

Methods

Line 24 State the community used

begin with…. A total of 392

Line 31 with a mean of 21.25 years (sd = 1.32)…. rewrite as Mean (�SD) ie.

21.25( �1.32)

Line 35 “on occasion” should be replaced with…occasionally.

Conclusion

BC should be defined on first use

Introduction

Line 53 delete “lying”

Line 55 is the sentence in reference to Ethiopia? If otherwise can author back up with references?

Line 61 “Young women's cancers are mostly diagnosed at an advanced stage” Could be rewritten as….. Young women with cancer are often diagnosed with advanced stage breast cancer.

Line 72 these references does not speak BSE in Ethiopia but rather other African countries.. Could a reference to that effect be included? Or sentence rephrased?

Materials and Methods

Line 89 period design…? What is that exactly

Line 99 study design: Information on study design is quite scanty I recommend a beef up. Or can be merged with study area… as study area and study design.

Line 103 were the young adults purposively selected? Please state so.

Line 125 The data were collected by 25 trained health extension workers under the supervision of five health officers and nurses…..Considering this information, I find it intriguing that this manuscript is a sole authorship one. Any comments on this?

Line 126… “that were closed households or had no participants were visited three times”. Should be rephrased sentence seem incomplete.

Line 133 “signals” ... change to “signs”

Line 142 BSE forms?

Line 188 the appropriate consent for participation should be a written informed consent.

Results

Line 193 Mean And SD data should be presented in standard formats

write as…. Mean (�SD)

Table 1 remove the year in the title.

If the questionnaire was self-administered, how was the 2.4% illiterate population assessed using the questionnaire? This must be elaborated in the methods.

Table 1 A 7.8% of married young adult in the study aea is worth a mention in the discussion.

Mean And SD data should be presented in standard formats through the manuscript

Percent should be written as Percent (%) to avoid the use of the symbol through the table

Academic status could be replaced with Educational Background.

The inclusion of the year in the table and figure titles should be revised.

Table 2 Percent should be accompanied by its Symbol.

Line 240.. title is not reflective of the information provided

Discussion

Line 273.. only one study is cited.

Line 275.. what is accounting for the poor knowledge in your study area?

Line 267 why so? Any suggestions to improve this outcome. That should a more precise focus of the discussion. Not merely stating what the data says.

Line 283 what is accounting for this low/poor knowledge BSE in south Ethiopia or your study population. Explain

Use consistent decimal points for all the percentage sated in the tables and discussion

Line 311 it is .. rather than it’s..

Line 317 “Confidence is supposed to come after knowledge”. Can be rewritten as….. “Knowledge builds confidence”

Line 323 what some of these barriers… as they are not mentioned clearly in the write up

Line 328 it is .. rather than it’s..

Line 332 I think that….. “The possible limitation of this work is that perceptions and behavioral practices of other breast cancer screening methods, such as clinical breast examination and mammography, were not assessed”. …. Is not the scope of the study and thus does not constitute a study limitation but may be a recommendation for additional studies. Do Reconsider!

Conclusion

Line 339 may not be necessary as part of the conclusion. Conclusion should be precise, concise, straight to the point.

Reviewer #2: 1. It is not clear why only women 20-24 were included in the study while older women are also eligible for BSE. This raises a question in the significance of the study? Otherwise the findings need to be put in context and discussed accordingly.

2. The title of the study seems as if the study is generalizable to Ethiopian women but only included women in one town. This is misleading and need to be corrected to mention the study locality in the title.

3. The samples size assumption of 20% is taken not from Ethiopian or African women. The SS is difficult to ensure adequacy. In such cases 50% should have been taken. In addition no justification given why design-effect if only 1.5 was taken while ti could have been 2. All these raise concerns on the adequacy of sample size.

4. The health belief model tool is a generic one. There is no mention of reliability and validity test for Ethiopian population. How can we rely in this measurement?

5. The data in Table 6 (association test) are not fully presented for most cells. In addition data only for those significantly associated is included. This is not appropriate. All variables irrespective of status if association were supposed to be presented.

6. PLOS authors have the option to publish the peer review history of their article (what does this mean?). If published, this will include your full peer review and any attached files.

Reviewer #1: No

Reviewer #2: No

---

## [Author Response · Author response to Decision Letter 0]

21 Jun 2022

PONE-D-22-07638

Perceptions and knowledge of breast cancer and breast self-examination among young adult Ethiopian women: application of the health belief model

PLOS ONE

Dear Dr. Assfa,

Thank you for submitting your manuscript to PLOS ONE. After careful consideration, we feel that it has merit but does not fully meet PLOS ONE’s publication criteria as it currently stands. Therefore, we invite you to submit a revised version of the manuscript that addresses the points raised during the review process.

If applicable, we recommend that you deposit your laboratory protocols in protocols.io to enhance the reproducibility of your results. Protocols.io assigns your protocol its own identifier (DOI) so that it can be cited independently in the future.

For instructions see: https://journals.plos.org/plosone/s/submission-guidelines#loc-laboratory-protocols. Additionally, PLOS ONE offers an option for publishing peer-reviewed Lab Protocol articles, which describe protocols hosted on protocols.io. Read more information on sharing protocolsathttps://plos.org/protocols?utm_medium=editorial email&utm_source=authorletters&utm_campaign=protocols.

We look forward to receiving your revised manuscript.

Kind regards,

Alvaro Galli

Academic Editor

PLOS ONE

 Journal Requirements:

Comments :1. Please ensure that your manuscript meets PLOS ONE's style requirements, including those for file naming. The PLOS ONE style templates can be found at https://journals.plos.org/plosone/s/file?id=wjVg/PLOSOne_formatting_sample_main_body.pdf &https://journals.plos.org/plosone/s/file?id=ba62/PLOSOne_formatting_sample_title_authors_affiliations.pdf

Responses#: Thank you for your comment. I formatted the manuscript in accordance with PLOS ONE's style guidelines and template.

Comments: 2. you indicated that you had ethical approval for your study. In your Methods section, please ensure you have also stated whether you obtained consent from parents or guardians of the minors included in the study or whether the research ethics committee or IRB specifically waived the need for their consent.

Responses#: Your suggestion is much appreciated. Participants in this study, however, were young adult women aged 20 to 24, who had reached the legal age to consent to participate in the study independently under Ethiopian law. As a result, no parental or guardian consent was requested.

Comments: Upon re-submitting your revised manuscript, please upload your study’s minimal underlying data set as either Supporting Information files or to a stable, public repository and include the relevant URLs, DOIs, or accession numbers within your revised cover letter. For a list of acceptable repositories, please see http://journals.plos.org/plosone/s/data-availability#loc-recommended-repositories. Any potentially identifying patient information must be fully anonymized.

Responses#: Thank you very much; I uploaded my data availability statement so that the minimum of data set is fully available in figshare public data repository (Doi=10.6084/m9.figshare.20103206 ,URLs= https://doi.org/10.6084/m9.figshare.20103206) 

 Important: If there are ethical or legal restrictions to sharing your data publicly, please explain these restrictions in detail. Please see our guidelines for more information on what we consider unacceptable restrictions to publicly sharing data: http://journals.plos.org/plosone/s/data-availability#loc-unacceptable-data-access-restrictions. Note that it is not acceptable for the authors to be the sole named individuals responsible for ensuring data access. We will update your Data Availability statement to reflect the information you provide in your cover letter.

Reviewers' comments: 

Reviewer's Responses to Questions

Comments to the Author

1. Is the manuscript technically sound, and do the data support the conclusions?

Reviewer #1: Partly

Reviewer #2: Yes

2. Has the statistical analysis been performed appropriately and rigorously?

Reviewer #1: Yes

Reviewer #2: Yes

Comments: 3. Have the authors made all data underlying the findings in their manuscript fully available?

Reviewer #1: No

Reviewer #2: No

Responses#: Thank you very much; I updated my data availability statement so that the minimum of data set is fully available in figshare public data repository (Doi=10.6084/m9.figshare.20103206 ,URLs= https://doi.org/10.6084/m9.figshare.20103206) .________________________________________

4. Is the manuscript presented in an intelligible fashion and written in standard English?

Reviewer #1: Yes

Reviewer #2: Yes

5. Review Comments to the Author

Reviewer #1: Perceptions and knowledge of breast cancer and breast self-examination among young adult Ethiopian women: application of the health belief model

REVIEW COMMENTS

The study was carried out to assess the knowledge and perceptions of young adult women in Southwest Ethiopia about breast cancer and breast self-examination (BSE). This is a good manuscript with very robust methods. The methodology has been comprehensibly and logically described for reproducibility. The write up and presentation good, however, some concerns need to be addressed. I recommend a major revision

Major Revision

Abstract

Methods

comments: Line 24 State the community used begin with…. A total of 392

Responses: I'm grateful to the reviewer who brought this to my attention. I have revised it. The changes made are indicated in track changed document. The changes are also found on the revised manuscript document [pages2, lines 24].

comments: Line 31 with a mean of 21.25 years (sd = 1.32)…. rewrite as Mean (�SD) ie.

21.25( �1.32)

Responses I agree and have updated line 31. The new sentence reads as follows “The respondents' ages ranged from 20 to 24, with a mean of 21.25 (� 1.32) years.” The changes made are indicated in track changed document. The changes are also found on the revised manuscript document [pages2, lines 30].

comments:Line 35 “on occasion” should be replaced with…occasionally

Response #: I agree and have changed “on occasion” to “occasionally”. The changes made are indicated in track changed document. The changes are also found on the revised manuscript document [pages2, lines 34].

Conclusion

comments: BC should be defined on first use

Response #: This is a valid observation. “BC” has been replaced with “breast cancer”. The changes made are indicated in track changed document. The changes are also found on the revised manuscript document [pages2, lines 39].

Introduction

comments: Line 53 delete “lying”

Response #: I agree and have removed the term "lying" from the sentence.

comments: Line 55 is the sentence in reference to Ethiopia? If otherwise can author back up with references?

Response #: Thank you for sharing your thoughts. I've provided references to back it up. The changes made are indicated in track changed document. The changes are also found on the revised manuscript document [pages3, lines 52].

comments: Line 61 “Young women's cancers are mostly diagnosed at an advanced stage” Could be rewritten as….. Young women with cancer are often diagnosed with advanced stage breast cancer. 

Response #: I agree with you, and I've rewritten it as follows: “Young women with cancer are often diagnosed with advanced stage breast cancer.” The changes made are indicated in track changed document. The changes are also found on the revised manuscript document [pages3, lines 58].

comments:Line 72 these references does not speak BSE in Ethiopia but rather other African countries. Could a reference to that effect be included? Or sentence rephrased?

Response #: This observation is correct. I have included a reference to Ethiopia’s BSE problem. The changes made are indicated in track changed document. The changes are also found on the revised manuscript document [pages4, lines 73].

Materials and Methods

comments:Line 89 period design…? What is that exactly

Response #: I am sorry for the error. “Study area, study period, and study design" was supposed to be the meaning. I've fixed the problem, and it's now written as “study area and study design”. The changes made are indicated in track changed document. The changes are also found on the revised manuscript document [pages5, lines 95].

comments:Line 99 study design: Information on study design is quite scanty I recommend a beef up. Or can be merged with study area… as study area and study design.

Response #: I welcome your insightful suggestion, and it has been combined with the study area and study design.

comments: Line 103 were the young adults purposively selected? Please state so.

Response #: Thank you for alerting me to this; however, I do not believe it is better to state so. Because I used a probability sampling approach rather than a purposive approach in the recruitment process. I hope you will keep me updated on your second version revision process if I don't get your view point right.

comments: Line 125 The data were collected by 25 trained health extension workers under the supervision of five health officers and nurses….Considering this information, I find it intriguing that this manuscript is a sole authorship one. Any comments on this?

Response #: Thank you for drawing my attention to this. I think it is important to briefly describe my data collectors who are the Health Extension Workers (HEW). According to Ethiopia's present health-care delivery system, each kebele (a smaller administrative unit) is allotted one or two HEW. They are dominantly female who expected to be well-versed in the community's culture and to be native speakers of the local language. They provide 16 different packages of basic health services. They also do a home visit and organize health and associated data, which is then documented in an organized manner in each family folder in their health post. In general, HEWs are valuable information sources with a broad spectrum of knowledge about their society. That's why I used them to gather my data. For each selected kebele, I used one HEW as responsible data collector to gather data in their respective kebele only without sending them to other kebele to keep their routine tasks on track as usual. That is why the data collectors’ number seems to be higher.

comments:Line 126… “that were closed households or had no participants were visited three times”. Should be rephrased sentence seem incomplete.

Response #: Thank you so much for your good insight, I have rephrased the sentence and the new sentence reads as follows: “Households that were either closed or lacked a respondent were visited three times.” The changes made are indicated in track changed document. The changes are also found on the revised manuscript document [pages7, lines 131].

comments: Line 133 “signals” ... change to “signs”

Response #: I agree with you, I have changed “signals” to “signs”. The changes made are indicated in track changed document. The changes are also found on the revised manuscript document [pages7, lines 139]. 

Comments: Line 142 BSE forms?

Response #: My apologies for using unfamiliar terminology, it was to mean a tool to measure BSE. I have rewritten it as “BSE knowledge measurement” The changes made are indicated in track changed document. The changes are also found on the revised manuscript document [pages7, lines 148].

comments:Line 188 the appropriate consent for participation should be a written informed consent.

Response #: I agree with you that written informed consent is preferable. Even if I use verbal consent, data collectors fully explain the study's purpose, anonymity, confidentiality, and voluntariness of participation before obtaining participants' consent or collecting data. 

Results

comments: Line 193 Mean and SD data should be presented in standard formats write as…. Mean (�SD) 

Response #: Thank you for your comment, I have revised and the result has been amended and the new sentence reads as follows: “The participants' ages ranged from 20 to 24, with a mean of 21.25(�1.32) years.” The changes made are indicated in track changed document. The changes are also found on the revised manuscript document [pages10, lines 199].

comments:Table 1 remove the year in the title. 

Response #: I agree with you, I have removed the ‘year” in the title of table 1 [pages10, lines 205].

comments:If the questionnaire was self-administered, how was the 2.4% illiterate population assessed using the questionnaire? This must be elaborated in the methods.

Response #: I appreciate the reviewer bringing this to my attention. In response to reviewers' suggestions, I revised and added additional elaboration to the data collection procedure. The changes made are indicated in track changed document. The changes are also found on the revised manuscript document [pages7, lines 134-138].

comments:Table 1 A 7.8% of married young adult in the study area is worth a mention in the discussion.

Response #: Thank you for taking the time to leave such an insightful comment. In the discussion section of the manuscript, I mentioned the marital status of the study participants. The modifications made are noted in the document's track changed section. The changes are reflected in the amended manuscript as well [pages 19, lines 316-323].

comments: Mean And SD data should be presented in standard formats through the manuscript

Response #: I agree with you and I have rewritten the Mean and SD in standard format [page 10, line 199]

comments: Percent should be written as Percent (%) to avoid the use of the symbol through the table

Response #: I agree with you and I have written the Percent as Percent (%) [ table 1 column 4 row1] and the symbols % in the table 1 have been removed [ table 1 column 4].

comments:Academic status could be replaced with Educational Background.

Response #: Thank you for the feedback I have replaced “Academic status” by “Educational Background” [table 1 column 1 row 4]

comments: The inclusion of the year in the table and figure titles should be revised.

Response #: Thank you for the suggestion, I have revised the year in all table and fig title [page 10, line 205] [page 11, line 216,217] [page 13, line 228,231] [page 14, line 238] [page 15, line 248] and [page 16, line 261,262].

comments: Table 2 Percent should be accompanied by its Symbol.

Response #: Thank you, Percent in able 2 has written as Percent (%) [page 11, table 2 column 4]

comments: Line 240.. title is not reflective of the information provided

Response #: Thank you for your feedback; if my understanding of your concern about line 240 is correct, it is a subheading given for young adult women's perception towards breast cancer and BSE, not the title of the table (table-4) above it. Young adult women's perceptions of breast cancer and BSE have been described under this subheading, which includes their perceived (susceptibility, severity, benefit, barriers, threat, outcome expectation or net benefit of BSE, and perceived confidence or self-efficacy). I have slightly modified the subheading. The changes made are indicated in track changed document. The changes are also found on the revised manuscript document [pages 15, lines 240].

Discussion

comments: Line 273. only one study is cited.

Response #: I agree and have updated with two additional references. The changes made are indicated in track changed document. The changes are also found on the revised manuscript document [pages 18, lines 284].

comments: Line 275.. what is accounting for the poor knowledge in your study area?

Response #: Thank you for the question. It is possible that this is due to the fact that cancer receives less attention [page 17,278-279]

comments: Line 267 why so? Any suggestions to improve this outcome. That should a more precise focus of the discussion. Not merely stating what the data says.

Response #: Thank you for taking the time to leave such an insightful comment. I've included possible contributing factors as well as recommendations for raising young women's awareness of breast health. The changes are added in the track changed manuscript document. They are also found on the revised manuscript document on [page 17, lines 278-280].

comments: Line 283 what is accounting for this low/poor knowledge BSE in south Ethiopia or your study population. Explain

Response #: Thank you for your valuable insight. I have added and discussed the possible contributing factors for respondents (young adult women) poor knowledge in my study area. The changes are added in the track changed manuscript document. They are also found on the revised manuscript document on [page 18, lines 300-303].

comments: Use consistent decimal points for all the percentage sated in the tables and discussion

Response #: I appreciate and thank you for your thoughtful remarks. I agree with your point of view. All fractions have been rounded to the nearest two decimal places for all the percentages sated in the paper. The changes are made in the track changed manuscript document

comments:Line 311 it is .. rather than it’s.

Response #: I have replaced the word “it’s” by “it is” [page 20, lines 333].

comments:Line 317 “Confidence is supposed to come after knowledge”. Can be rewritten as….. “Knowledge builds confidence” 

Response #: I accepted your suggestion and have rewritten as “Knowledge builds confidence” The changes are added in the track changed manuscript document. They are also found on the revised/amended manuscript document on [page 20, lines 339-340].

comments: Line 323 what some of these barriers… as they are not mentioned clearly in the write up

Response #: Thank you for your question and suggestion. In this study perceived barriers of BSE were measured using 7 items stated as: (1) BSE is embarrassing to me, (2) BSE takes too much time, (3) It is hard to remember to do breast examination, (4) I am afraid I would not be able to do breast self-exams,(5) I don't have enough privacy to do breast examination, (6) BSE is not necessary if you have a breast exam by a healthcare provider and (7) I have other problems more important than doing BSE. Overall, lack of knowledge, skill and privacy to perform BSE are some of the barriers. Apart from that, believing that BSE embracing, and time-consuming activity which is also related with lower knowledge, giving less priority for them practices compared to another problem which they believe important than BSE. The changes are added in the track changed manuscript document. They are also found on the revised/amended manuscript document on [page 20, Line 345-346].

comments:Line 328 it is .. rather than it’s..

Response #: Thank you, I have replaced the word “it’s” by “it is” [ page 21, Line 350].

comments: Line 332 I think that….. “The possible limitation of this work is that perceptions and behavioral practices of other breast cancer screening methods, such as clinical breast examination and mammography, were not assessed”. …. Is not the scope of the study and thus does not constitute a study limitation but may be a recommendation for additional studies. Do Reconsider!

Response #: Thank you for your valuable suggestion. The statement “The possible limitation of this work is that perceptions and behavioral practices of other breast cancer screening methods, such as clinical breast examination and mammography, were not assessed” has been deleted. The changes are added in the track changed manuscript document.

Conclusion

comments: Line 339 may not be necessary as part of the conclusion. Conclusion should be precise, concise, straight to the point.

Response #: The statement in conclusion “In this study, health beliefs and reported breast self-examination practices of young adult women were assessed.” has been removed as suggested by reviewer. The changes are added in the track changed manuscript document.

Reviewer #2: 

comments:1. It is not clear why only women 20-24 were included in the study while older women are also eligible for BSE. This raises a question in the significance of the study? Otherwise, the findings need to be put in context and discussed accordingly

Response #: I appreciated and thank you for your valuable insight. I have provided additional points to answer why only young adult women (20-24 years old) were included in my study. The change has been highlighted in the track changed manuscript documented. It is found on [page 3 lines number 58-61], [page 4, line number 74-76] and [page 4, line number 81-83] in the revised manuscript document.

comments:2. The title of the study seems as if the study is generalizable to Ethiopian women but only included women in one town. This is misleading and need to be corrected to mention the study locality in the title

Response #: Thank you for your concern and question. I agree with you, thus I've slightly modified the title to look like: “Perceptions and knowledge of breast cancer and breast self-examination among young adult women in South west Ethiopia: application of the health belief model”. The change has been highlighted in the track changed manuscript documented and it is found on [page 1, lines number 2] in the revised manuscript document.

comments:3. The samples size assumption of 20% is taken not from Ethiopian or African women. The SS is difficult to ensure adequacy. In such cases 50% should have been taken. In addition, no justification given why design-effect if only 1.5 was taken while ti could have been 2. All these raise concerns on the adequacy of sample size.

Response #: I appreciate the reviewer who brought this to my attention. I totally agree that the safest choice for obtaining the largest sample size is to use a population proportion of 50%. When it comes to the design effect, researchers adopt a default value of 1.5 to 2.0 when determining sample size because the design effect (deff) is typically regarded as unknown before to a survey. To account for the expense, workload, and time required to collect data from a geographically scattered study population, I utilized the design effect 1.5 and p=0.203 in my sample.

comments:4. The health belief model tool is a generic one. There is no mention of reliability and validity test for Ethiopian population. How can we rely in this measurement?

Response #: I agree with you and thank you for your concern. I've reported the Cronbach alpha as a measure of consistency[page 8, L 163-164]. An expert panel was consulted to ensure face validity when the tool's validity was examined. 

comments:5. The data in Table 6 (association test) are not fully presented for most cells. In addition, data only for those significantly associated is included. This is not appropriate. All variables irrespective of status if association were supposed to be presented.

Response #: Thank you for taking the time to provide me with your valuable input. A change has been made to the table. If my understanding is correct, the table contained all variables, independent of their significant status. Continuous independent variables were denoted by the letter “a” in superscript style, and their cells remained vacant. The change has been highlighted in the track changed manuscript documented and it is found on [page 17, line number 262-263] and [table 6 column 1 row 9-11] in the revised manuscript document.________________________________________

6. PLOS authors have the option to publish the peer review history of their article (what does this mean?). If published, this will include your full peer review and any attached files.

Do you want your identity to be public for this peer review? For information about this choice, including consent withdrawal, please see our Privacy Policy.

Reviewer #1: No

Reviewer #2: No

Thank you

The Author

---

## [Decision Letter · Decision Letter 1]

12 Aug 2022

PONE-D-22-07638R1Perceptions and knowledge of breast cancer and breast self-examination among young adult women in South west Ethiopia: application of the health belief modelPLOS ONE

Dear Dr. Assfa,

Thank you for submitting your manuscript to PLOS ONE. After careful consideration, we feel that it has merit but does not fully meet PLOS ONE’s publication criteria as it currently stands. Therefore, we invite you to submit a revised version of the manuscript that addresses the points raised during the review process.

We look forward to receiving your revised manuscript.

Kind regards,

Alvaro Galli

Academic Editor

PLOS ONE

Journal Requirements:

Reviewers' comments:

Reviewer's Responses to Questions

**Comments to the Author**

1. If the authors have adequately addressed your comments raised in a previous round of review and you feel that this manuscript is now acceptable for publication, you may indicate that here to bypass the “Comments to the Author” section, enter your conflict of interest statement in the “Confidential to Editor” section, and submit your "Accept" recommendation.

Reviewer #1: (No Response)

2. Is the manuscript technically sound, and do the data support the conclusions?

Reviewer #1: Yes

3. Has the statistical analysis been performed appropriately and rigorously? 

Reviewer #1: Yes

4. Have the authors made all data underlying the findings in their manuscript fully available?

Reviewer #1: Yes

5. Is the manuscript presented in an intelligible fashion and written in standard English?

Reviewer #1: Yes

6. Review Comments to the Author

Reviewer #1: While most of the comment raised have been addressed in a few needs to be looked into.

Abstract

Methods

Line 24 which community Again, I suggest you State the community employed for the study.

Line 48… replace occurred with occurs

Line 58.. again look at line 58.. the repetition of young women in the 2 sentences is not appropriate and should be modified.

Line 134… references for where questionnaire was adapted from, should be stated.

Line 139 use society… not society’s

Line 194 Again… the appropriate consent for participation should be a written informed consent not verbal agreement. This an important part of the study.

Line 216 and 217 … why do you have titles for tables and figures? Why present the same information in Tables and Figures. Explain please.

Line 228 should read….. Respondents knowledge about early warning signs of breast cancer, Gurage zone

In table 3… sd.. should be SD….. out of 11?explain please.

Line 292 A stronger justification for difference is imperative.

Line 294 delete ..participated

Line 354… rewrite as…..only young adult women participated in the study.

Line 359… why threats?

7. PLOS authors have the option to publish the peer review history of their article (what does this mean?). If published, this will include your full peer review and any attached files.

Reviewer #1: No

---

## [Author Response · Author response to Decision Letter 1]

17 Aug 2022

August, 2022

From: Authors

To: PLOS ONE editorials 

Subject: Response to reviewers

Dear Editor and reviewers:

I would like to thank the editors and reviewers for their time, careful reading, and thorough and helpful remarks and questions about my manuscript, as well as the opportunity to improve and resubmit it.

I believe that my revised paper has substantially improved as a result of the reviewers’ remarks. I am glad to submit the revised research manuscript for consideration in PLOS ONE. 

 I responded to each reviewer's questions, comments, and suggested revisions one by one. I used the same letter to write my responses after copying the review decision from the editorial manager's submission menu. Editors' and reviewers' questions and comments are italicized, and my responses to the reviewers' comments are presented in plain text immediately below them. All of the modifications I made in the revised manuscript were referenced by page and line numbers. Please, follow my responses in a yellow mark to every comment or question in a green mark. 

Once again, thank you for your time and attention to my paper, and I eagerly await your response. 

Sincerely!

Kenzudin Assfa Mossa (corresponding author)

Wolkite University 

kenzaheri@gmail.com

+251923702054

PONE-D-22-07638R1

Perceptions and knowledge of breast cancer and breast self-examination among young adult women in southwest Ethiopia: application of the health belief model

PLOS ONE

Dear Dr. Assfa,

Thank you for submitting your manuscript to PLOS ONE. After careful consideration, we feel that it has merit but does not fully meet PLOS ONE’s publication criteria as it currently stands. Therefore, we invite you to submit a revised version of the manuscript that addresses the points raised during the review process.

• A rebuttal letter that responds to each point raised by the academic editor and reviewer(s). You should upload this letter as a separate file labeled 'Response to Reviewers'

We look forward to receiving your revised manuscript.

Kind regards,

Alvaro Galli

Academic Editor

PLOS ONE

Journal Requirements:

Comments: Please review your reference list to ensure that it is complete and correct. If you have cited papers that have been retracted, please include the rationale for doing so in the manuscript text, or remove these references and replace them with relevant current references. Any changes to the reference list should be mentioned in the rebuttal letter that accompanies your revised manuscript. If you need to cite a retracted article, indicate the article’s retracted status in the References list and also include a citation and full reference for the retraction notice.

Response: I appreciate the suggestions. I checked the reference list; it is complete, correct, and free of any references that have been retracted. Scite (https://scite.ai/users/kenza-abazinab-pn5Ej/reference-checks) revealed no references that had been retracted or that had editorial concerns. In addition, I haven't changed the reference list.

Reviewers' comments:

Reviewer's Responses to Questions

Comments to the Author

1. If the authors have adequately addressed your comments raised in a previous round of review and you feel that this manuscript is now acceptable for publication, you may indicate that here to bypass the “Comments to the Author” section, enter your conflict-of-interest statement in the “Confidential to Editor” section, and submit your "Accept" recommendation.

Reviewer #1: (No Response)

2. Is the manuscript technically sound, and do the data support the conclusions?

Reviewer #1: Yes

3. Has the statistical analysis been performed appropriately and rigorously?

Reviewer #1: Yes

4. Have the authors made all data underlying the findings in their manuscript fully available?

Reviewer #1: Yes

5. Is the manuscript presented in an intelligible fashion and written in standard English?

Reviewer #1: Yes

6. Review Comments to the Author

Reviewer #1: While most of the comment raised have been addressed in a few needs to be looked into.

Abstract

Methods

Comments: Line 24 which community Again, I suggest you State the community employed for the study.

Response: I'm appreciative to the reviewer who made me aware of this. It has been updated and the new sentence read as: “A community-based cross-sectional study was carried out in Gurage zone, southwest Ethiopia, in 2021.”. The track changed document shows the modifications that have been made. Pages 2 and line 24 of the revised manuscript document contain the modifications.

Comments: Line 48… replace occurred with occurs

Response: I agree and the term occurred replaced by occurs. The changes made are indicated in track changed document. The changes are also found on the revised manuscript document [pages3, lines 49].

Comments: Line 58.. again look at line 58.. the repetition of young women in the 2 sentences is not appropriate and should be modified.

Response: Thank you for the insightful comments. The pronoun "they" was used in place of "young adult women" in the second sentence reads as: "They have a variety of issues, including a higher likelihood of biologically aggressive illness and metastatic disease at diagnosis, which results in a worse prognosis, more aggressive treatments and long-term treatment-related toxicities, and unique psychosocial concerns". The track changed document shows the modifications that have been made. Pages 3 and lines 59 of the revised manuscript document contain the modifications.

Comments: Line 134… references for where questionnaire was adapted from, should be stated.

Response: This is a valid observation. I have added references for where questionnaire was adapted from. The changes made are indicated in track changed document. The changes are also found on the revised manuscript document [pages7, lines 135-136].

Comments: Line 139 use society… not society’s

Response: I agree, and I've changed the word "society's" in the sentence with the word society. Pages 7 and line 140 of the revised manuscript document contain the modifications.

Comments: Line 194 Again… the appropriate consent for participation should be a written informed consent not verbal agreement. This an important part of the study.

Response: Thank you for sharing your thoughts. I have revised it. The new sentence read as follow: “The purpose of the study was communicated to the respondents, and signed consent from each participant was obtained prior to data collection." The changes made are indicated in track changed document. The changes are also found on the revised manuscript document [page10, lines 195-197].

Comments: Line 216 and 217 … why do you have titles for tables and figures? Why present the same information in Tables and Figures. Explain please.

Response: Your thoughtful concern is appreciated. The difference between the data shown in table 2 and fig. 1 is that the table pertains to the frequency distribution of responses given to each of the 11 items used to measure respondents' knowledge of breast cancer risk factors. In response to your suggestion, I eventually combined the data from Fig 1 into the table 2 and eliminated the Fig. The changes made are indicated in track changed document and found on the revised manuscript document [pages11, lines 215-217]. To overall knowledge of respondents about breast cancer risk factors categorized into three categories: low, moderate, and high. The frequency distribution is shown in the last three rows of table 2, page 12 of the revised manuscript.

Comments: Line 228 should read….. Respondents knowledge about early warning signs of breast cancer, Gurage zone

Response: Thank you so much for your comment, I have rephrased the title and the new sentence reads as: “Respondents knowledge about early warning signs of breast cancer, Gurage zone”. The changes made are indicated in track changed document. The changes are also found on the revised manuscript document [pages13, lines 229]. 

Comments: In table 3… sd.. should be SD….. out of 11?explain please.

Response: I agree with you, I have changed “sd” to “SD”. 11 is the sum score of the 11 items I used to generate the composite score of respondents' warning sign-related knowledge of breast cancer. The revised manuscript document (pages 13 and table 3) and the track changed document both indicate the changes that were made.

Comments: Line 292 A stronger justification for difference is imperative.

Response: thank you so much for valuable suggestions, I have added the possible factors that could contributed the reported difference in knowledge. the new sentences read as: “Since older and married women are more interested in and aware of BC (20,37,38), this difference could be explained by our younger participants (20–24 years of age), who were younger than those in Beirut (18–65 years), and whose proportion of married people (7.77%) was significantly lower than Beirut's (49.5%) (36)”. The changes made are indicated in track changed document and also found on the revised manuscript document [pages18, lines 292-296]. 

Comments: Line 294 delete ..participated

Response: I agree with your suggestion, and the term “participated” removed from the sentence. The changes made are indicated in track changed document.

Comments: Line 354… rewrite as…..only young adult women participated in the study.

Response: Thank you for your comment, I have revised it and the new sentence reads as: “Since only young adult women participated in the study, it does not reflect the perceptions and knowledge of young adult males.” The changes made are indicated in track changed document. The changes are also found on the revised manuscript document [pages21, lines 358-359].

Comments: Line 359… why threats?

Response: I appreciate the query. It is well established that a slight or moderate threat plays a big part in making people worry about certain health issues, such as breast cancer. According to the findings of this study, which are shown in table 5, a significant number (96%) of survey participants were less concerned about breast cancer. Because of this it has been included in the conclusion.

7. PLOS authors have the option to publish the peer review history of their article (what does this mean?). If published, this will include your full peer review and any attached files.

Do you want your identity to be public for this peer review? For information about this choice, including consent withdrawal, please see our Privacy Policy.

Reviewer #1: No

Thank you

The Author

---

## [Editor Report · Decision Letter 2]

8 Sep 2022

Perceptions and knowledge of breast cancer and breast self-examination among young adult women in South west Ethiopia: application of the health belief model

PONE-D-22-07638R2

Dear Dr. Assfa,

We’re pleased to inform you that your manuscript has been judged scientifically suitable for publication and will be formally accepted for publication once it meets all outstanding technical requirements.

Kind regards,

Alvaro Galli

Academic Editor

PLOS ONE

Additional Editor Comments (optional):

All the comments have been addressed and changes added; in the current form (Revision number 2) the manscript is accepted for publication.
---

## [Editor Report · Acceptance letter]

12 Sep 2022

PONE-D-22-07638R2 

Perceptions and knowledge of breast cancer and breast self-examination among young adult women in southwest Ethiopia: application of the health belief model 

Dear Dr. Assfa Mossa:

I'm pleased to inform you that your manuscript has been deemed suitable for publication in PLOS ONE. Congratulations! Your manuscript is now with our production department. 

Kind regards, 

on behalf of

Dr. Alvaro Galli 

Academic Editor

PLOS ONE